# Functional Improvement and Satisfaction with a Wearable Hip Exoskeleton in Community-Living Adults

**DOI:** 10.3390/healthcare11050643

**Published:** 2023-02-22

**Authors:** Jihye Kim, Hyelim Chun, Su-Hyun Lee, Hwang-Jae Lee, Yun-Hee Kim

**Affiliations:** 1Center for Prevention and Rehabilitation, Samsung Medical Center, Department of Physical and Rehabilitation Medicine, Heart Vascular Stroke Institute, Sungkyunkwan University School of Medicine, Seoul 06351, Republic of Korea; 2Robot Business Team, Samsung Electronics, Suwon 16499, Republic of Korea; 3Department of Health Sciences and Technology, Department of Medical Device Management and Research, Department of Digital Health, SAIHST, Sungkyunkwan University, Seoul 06351, Republic of Korea

**Keywords:** exoskeleton device, gait, robot, personal satisfaction, physical fitness

## Abstract

Demand for wearable devices and supportive technology is growing as these devices have the potential to enhance physical function and quality of life in users. The purpose of this study was to investigate usability and satisfaction after performing functional and gait exercise with a wearable hip exoskeleton in community-living adults. A total of 225 adults residing in the local community participated in this study. All participants performed 40 min of exercise once with a wearable hip exoskeleton in various environments. The EX1, which functions as a wearable hip exoskeleton, was used. Physical function was assessed before and after exercise with the EX1. After completing exercise with the EX1, the usability and satisfaction questionnaires were evaluated. Gait speed, timed up and go test (TUG), and four square step test (FSST) showed statistically significant improvements after exercise with the EX1 in both groups (*p* < 0.05). In the 6 min walking test (6MWT), a significant increase was observed in the middle-aged group (*p* < 0.05). In the short physical performance battery (SPPB), there was a significant improvement in the old-aged group (*p* < 0.05). On the other hand, positive results in usability and satisfaction were noticed in both groups. These results demonstrate that a single session of exercise with the EX1 was effective in improving physical performance of both middle- and old-aged adults, with positive feedback from most of the participants.

## 1. Introduction

Physical activity, which is associated with quality of life, decreases with age due to loss of muscle mass and strength [1], cardiac and respiratory disease [2], and other factors. Decreased physical activity during the aging process leads to metabolic disorders and other chronic diseases, such as cancer, diabetes, and cerebrovascular and cardiovascular diseases [3]. Regular physical activity such as walking, cycling, dancing, and running leads to positive physiological changes. Regular physical activity improves autonomic balance, bone density, capillary density, muscle fiber size, neuromuscular coordination, stroke volume, blood coagulation, and inflammation [4]. Moreover, these physiological changes result in decreased depression, weight gain, fractures, injurious falls, osteoporosis, and mortality, as well as increased physical and cognitive function [4]. A previous study reported that regular physical activity could increase the life expectancy of the world’s population [5].

Walking is one of the easiest regular physical activities. Walking is the cheapest and easiest activity for health promotion and consumes larger amounts of energy than other daily sports activities. Gait function refers to the mobility required for daily life performance, which can predict clinical conditions in various aspects [6]. In previous studies, inactive adults experienced 3–8% muscle loss over 10 years with decreased metabolic rate and fat accumulation during rest [7], but walking exercise improved walking economy and functionality [8], muscle size and quality [9], functional balance, and reaction time [10], which are related to gait quality.

Recently, functional exercise has been carried out under various conditions, such as applying a load, using props, or combining various conditions. Resistance gait exercise enhances balance and gait parameters [11] and decreases the fear of falling [12]. A previous study showed that, after resistance gait exercise using an underwater treadmill, gait parameters including step length, velocity, and cadence were significantly increased [13]. In addition, an assistive therapeutic exercise program is effective in improving preferred gait speed [14].

Robot technology in the healthcare field is being actively used in clinical practice as it has been developed in the surgical and rehabilitation medical fields. However, in recent years, the technological advances in daily assistive robots have gone beyond their limited use for patients in the medical field [15]. Wearable exoskeletons are being actively developed to assist and strengthen physical functions for not only disabled, but also non-disabled people. A wearable exoskeleton helps to improve gait by assisting with voluntary movement of the lower extremity joints during gait [16,17]. Furthermore, gait exercise with a wearable hip exoskeleton improves cardiopulmonary function. In a previous study, gait training with an assisted-exoskeleton robot required less oxygen consumption than a home exercise program of self-paced overground walking without a robot at the same speed in the elderly [18].

The EX1, developed by Samsung Electronics (Suwon, Republic of Korea), is a personalized robot with a light weight of 2.1 kg that is worn on the hip joints. In our previous study in the elderly, there were statistically significant improvements in gait speed, excessive muscle activity, respiration, and metabolic energy during gait with the EX1 compared to walking without it [16].

Demand for wearable devices and supportive technology is growing because these devices have potential to enhance physical functions and quality of life of users. Furthermore, among the factors to be considered when using a wearable robot, user stability and fit are important. Previous studies on the EX1 investigated the effect on physical function rather than usability and satisfaction with the wearable robot. Thus, wearable devices need to be evaluated by actual users in terms of wearability [19]. This study was designed to investigate the usability, satisfaction, and physical function from a single session of functional and gait exercise with the EX1 in community-living middle- and old-aged adults.

## 2. Materials and Methods

### 2.1. Study Participants

The participants were recruited from local community residents who use silver town or welfare centers. Two-hundred and twenty-five adults who met the following inclusion criteria were enrolled in this study: no history of central nervous system disease and age between 40 and 84 years. Subjects with uncontrolled severe high blood pressure or diabetes, history of uncontrolled cardiovascular disease, severe dizziness that might lead to a fall, and cognitive disorders that hinder the ability to understand or comply with study instructions were excluded. General characteristics of the subjects are summarized in Table 1. The study protocol was approved by the Institutional Review Board of Samsung Medical Center, Seoul, Korea (No. 2021-04-058), and informed consent was provided by all subjects before participating in the study.

### 2.2. Experimental Protocol

This study protocol was designed as a single group, and all participants performed a single session of exercise with EX1. All participants received 40 min of exercise with the EX1: 20 min of functional exercise including sit-to-stand and balance exercise and 20 min of gait exercise including stair climbing and over-ground and incline walking with the assist and resistance modes of EX1. To confirm the effect of a single session of exercise with EX1, physical functions measured by the 10 m walking test (10MWT), timed up and go test (TUG), four square step test (FSST), 6 min walking test (6MWT), and short physical performance battery (SPPB) were evaluated before and after the exercise program. All outcome measures were performed without the EX1. After completing a single session of exercise with the EX1, the usability and satisfaction questionnaire for EX1 were evaluated (Figure 1).

### 2.3. Wearable Hip Exoskeleton (EX1)

The EX1, which was developed by Samsung Electronics, is a minimized exoskeleton worn on the hip joints. It is very lightweight and user-customizable, weighing approximately 2.1 kg (Figure 2). EX1 can provide assistive or resistive torque forces around both hip joints for both extension and flexion direction during gait as needed, and it promotes physical function in daily life. EX1 consists of a pair of actuators that generate force on the left and right hip joints, a hip brace on the waist, a pair of thigh frames, and a thigh belt [20].

### 2.4. Measurement Tools

#### 2.4.1. Physical Function Evaluation

To measure physical function, 10MWT, TUG, FSST, 6MWT, and SPPB were performed. The 10MWT is the most effective method for predicting falls and evaluating gait ability by measuring gait speed, and it has high test–retest reliability in healthy adults (intra- and inter-tester reliability (ICC) = 0.93–0.91) [21]. For measurement, subjects were asked to walk a total of 15 m at a comfortable speed. The time it took to walk 10 m was measured, excluding the initial 2.5 m acceleration and the final 2.5 m deceleration, and the result in seconds was converted into speed (m/s). Measurements were repeated twice, and the average value was used.

To evaluate the dynamic balance ability among physical functions, TUG and FSST were measured. The TUG test is used as a standard test method in clinical practice as a representative test method for measuring gait ability as well as dynamic balance ability of elderly and brain injury patients, and it has high ICC in the elderly (ICC = 0.92–0.99) [22]. The FSST is a method of evaluating dynamic balance and movement ability. It measures the time it takes to walk forward, backward, and sideways over a low obstacle as fast as subjects can, and it has high test–retest reliability (ICC = 0.87) [23].

To evaluate walking endurance, 6MWT was measured. The 6MWT is a representative method for measuring gait aerobics and endurance in the elderly and patients with cardiopulmonary disease. To measure gait endurance, subjects are asked to walk a set trajectory for 6 min, and the total walking distance is measured. It has excellent test–retest reliability for the elderly (ICC = 0.95) [22].

SPPB is a test that consists of walking speed, sit-to-stand on a chair, and balance tests. It has been used as a predictive tool for dysfunction and can be helpful in functional monitoring in the elderly. The score ranges from a minimum of 0 points to a maximum of 12 points. SPPB has been shown to have validity as a predictor of fall risk and mortality and has excellent test–retest reliability for the elderly (ICC = 0.91) [24].

#### 2.4.2. The Usability and Satisfaction Questionnaire

As there was no appropriate usability and satisfaction questionnaire for evaluating the wearable hip exoskeleton, the questionnaire was developed and applied through previous research [17] and consultation with experts (Table 2). A questionnaire was developed by setting usability and satisfaction evaluation areas. The safety questionnaire is one of the most important parts to determine whether EX1 can be used safely. The operability question is a factor to realize the function of the gait robot. The questionnaire includes operational convenience, effectiveness, and efficiency.

### 2.5. Data Preprocessing and Statistical Analysis

All data were analyzed with SPSS version 22.0 program (IBM, Armonk, NY, USA). Results were calculated as mean and standard deviation values. Statistical significance levels for all measurements were set as *p* < 0.05.

Physical function evaluation, usability, and satisfaction tests were analyzed by dividing the age into two subgroups: a middle-aged group 40 to 64 years old and an old-aged group 65 to 84 years old. To evaluate the feasibility of a single session with the EX1, paired t-tests were used to compare outcome measures between pre- and post-exercise.

To evaluate the usability and satisfaction questionnaire, frequency analysis was used. In addition, the usability and satisfaction tests with standardized β coefficients in linear regression analysis were used to investigate the relationship between physical function and questionnaire.

## 3. Results

### 3.1. Physical Function

In the 10MWT, TUG, and FSST, significant increases after a single exercise session with the EX1 were observed in both groups (*p* < 0.05) (Figure 3). In the 6MWT, there was a statistically significant improvement only in the middle-aged group (*p* < 0.05). In the SPPB, there was a significant improvement in the old-aged group (*p* < 0.05) but not in the middle-aged group.

### 3.2. Usability and Satisfaction of the EX1

In the safety domain (involving items such as risk of falls and control of risk factors), after using the EX1, positive responses were confirmed in both groups. The responses were mostly positive in the satisfaction domain, but there were some items with difference by age group. The middle-aged group showed more positive responses than the old-aged group in the easiness of use, usefulness, and perception of the EX1. In addition, the middle-aged group showed more positive responses than the old-aged group in the questionnaire on whether the assist and resist mode of the EX1 helped with gait exercises. Though the old-aged group tended to prefer the assist mode, the middle-aged group tended to prefer the resistance mode of the EX1 (Figure 4).

### 3.3. Regression Analysis between General Characteristics, Usability, and Satisfaction

As a result of regression analysis of 128 cases with a combination of 16 dependent variables and 8 independent variables, there were 15 significant results (Y = aX + b) (Table 2). The dependent variable (Y) consisted of a total of 16 items related to safety (4 items) and satisfaction (12 items), and the independent variable (X) consisted of a total of 8 items: age, gender, height, weight, body mass index (BMI), health status (2 items; health conditions, level of activity), and experience of fall.

The younger the user, the easier they tend to think the EX1 was to use, they tend to think that the assist mode of the EX1 helps most with gait, and they tend to be willing to continue using the EX1. If the users think they have better health conditions, they tend to think the EX1 was easy to use. If the user is more active, they think that they control the risk posed by the EX1 (Appendix A).

### 3.4. Regression Analysis between Usability, Satisfaction, and Physical Function

As a result of regression analysis of 240 cases by combining 10 dependent variables and 24 independent variables, there were 25 significant results (Y = aX + b) (Table 2). The dependent variable (Y) consisted of a total of 10 items regarding functional evaluation baseline (5 items) and difference of functional evaluation between baseline and after single-session exercise with the EX1 (5 items). The independent variable (X) consisted of a total of 24 items related to age, gender, height, weight, body mass index (BMI), health status (2 items; health condition, level of activity), experience of fall, safety (4 items), and satisfaction (12 items).

If the users thought they have better health conditions, they tended to have a faster FSST and TUG. If the users considered themselves active, they tended to have faster gait speed, and walk farther in 6 min. If the users had experience of falling, they tended to have a lower SPPB, slower gait speed, and walk less in 6 min; after gait exercise with the EX1, these users tended to have a higher SPPB than before exercise.

If the users thought there was no risk of falling when they were turning or leaning forward while wearing the robot, after gait exercise with the EX1, they tended to have a lower SPPB. If the users thought the robot was easy to use, they tend to have a higher SPPB, and faster TUG; after gait exercise with the EX1, these users tended to have a faster FSST than before exercise (Appendix A).

## 4. Discussion

Our study demonstrates that a single session of exercise with the EX1 improved physical functions. In addition, the positive results of the EX1 were confirmed by conducting a usability and satisfaction survey after exercise with the EX1.

In this study, statistically significant improvements in gait speed, balance ability, and gait endurance were confirmed through a single session of exercise with the EX1. The results of this study suggest that a single session of exercise with the EX1 has several key advantages for physical function and efficiency. Functional exercise that is effective for improving gait function includes gait exercise at various speeds and directions, treadmill gait, and stair climbing. In addition, the resistance and assistance of the EX1 can be adjusted based on individual physical ability. Therefore, it improves physical function gradually by controlling the intensity and duration of the exercise.

To determine gait quality, we used objective, sensitive, and powerful measurement tools that measure gait performance, dynamic balance ability, and gait endurance [25]. In our study, there was a statistically significant improvement in gait speed. Previous studies have shown that lower gait speed is associated with age, education, and particularly modifiable factors such as impairment of activities of daily life, physical inactivity, and cardiovascular disease [26]. Gait speed is a clinical indicator related to survival rate and predicts functional ability in the elderly [27]. In a previous study, it was reported that gait speed increased by 0.1 m/s as the survival rate increased [28]. These results indicate the importance of staying active and healthy for middle- and old-aged people.

Balance ability, an essential factor in gait ability, is the ability to maintain balance against numerical movement and external stimuli, which reduces muscle weakness due to decreased physical activity and is highly associated with risk of fall due to dynamic balance ability during gait [29,30]. Physical changes due to aging lead to a decrease in gait function by reducing balance, and it is highly related to the incidence of injuries in the elderly [31]. In our study, a statistically significant improvement was shown in dynamic balance ability through a single session of exercise with the EX1. Falling is a serious problem that threatens the health of the elderly and can lead to premature death due to physical damage, psychological dysfunction, and onset of various diseases [32]. Preventing falls and improving body function require planned and consistent exercise. Decreased physical activity leads to muscle strength weakness, which causes reduced balance ability. Balance training is an essential element in an exercise program because it is highly correlated with risk of falls according to dynamic balance ability while walking.

Gait endurance is also one of the major factors influencing risk of falls in the elderly. It is known that the weaker the walking endurance, the higher the likelihood of a fall in the elderly. In our study, endurance improved immediately after a single session of exercise with the EX1 in both age groups, but a significant result was found only in the middle-aged group. Endurance is increased through long-term gait training [33], but in our study, the time was insufficient to derive significant results with a single session of exercise with the EX1. However, although it was not significant after a single session of exercise, walking distance for 6 min improved, indicating the potential for positive results with long-term exercise.

The SPPB is a fast and useful measurement tool for predicting falls [34]. In our study, the SPPB improved immediately in the old-aged group after a single session of exercise with the EX1, but no significant result was shown due to the ceiling effect in the middle-aged group.

This study was performed to confirm the usability and satisfaction of the EX1. Considering user physical condition and environment, it was divided into user questionnaire evaluation and user function evaluation, and the safety, operability, and satisfaction of the EX1 were evaluated. If an expert or designer conducts a usability evaluation while listening to users’ opinions, it is possible to understand the users’ needs, inconveniences, strengths, and expectations in a complex way. Regression analysis showed a difference in the experience feedback for the EX1 by age or physical function. However, there were no implications for usability, perceptions, or satisfaction with general characteristics other than age.

As the results of the questionnaire included usability and satisfaction, both middle- and old-aged groups had positive experience feedback for the EX1. Among them, more positive results were shown in the middle-aged group than the old-aged group in a few items. We think that these results were caused by middle-aged people being more open-minded and faster to learn new technologies than the elderly. In the questionnaire, both the assist and resistance modes of the EX1 helped with gait, with strongly positive answers in the middle-aged group compared to the old-aged group. In addition, in a questionnaire on the preference between the assist and resistance modes of the EX1, the old-aged group preferred the assist mode, but the middle-aged group preferred the resistance mode. These results indicated that the elderly, who have a decline in gait ability, prefer gait assist, but middle-aged people who need gait training prefer the resist mode of the EX1.

In regression analysis between general characteristics and questionnaire, users tended to think that using the EX1 would have no negative social perceptions if they thought they had lower health conditions. Younger users tend to have positive opinions of the EX1. Although there might be perceptions that exercise using robots is applied only to patients or the elderly [35,36], this study confirmed that healthy people had positive perceptions of exercise with robots. Therefore, we think that the EX1 can provide a meaningful exercise program not only for the elderly, but also for young people. In regression analysis between questionnaire and physical function, people who had experienced a fall within the last 6 months had better balance ability and physical function after a single session of exercise with the EX1. It was confirmed that the higher the satisfaction with the function of the robot, the better the physical endurance and dynamic balance. We think that such exercise will bring positive results when applied to people with reduced balance ability, which is a risk factor for falls.

## 5. Conclusions

This study demonstrated that a single session of exercise with the EX1 in middle- and old-aged persons improved physical performance, including gait and balance, and received positive feedback. A newly developed wearable hip exoskeleton, the EX1, is a potentially useful exercise device for improving gait and physical function not only in the elderly, but also in middle-aged people. In this study, we emphasized short-term usability and satisfaction evaluation of the EX1 for healthy subjects. In future studies, the long-term effects of the EX1, not only in healthy subjects but also in persons with impaired physical function, must be addressed. Additionally, it is necessary to conduct more structured studies and repeat pre–post experiments by repeating specific training protocols in scheduled programs.

## Figures and Tables

**Figure 1 healthcare-11-00643-f001:**
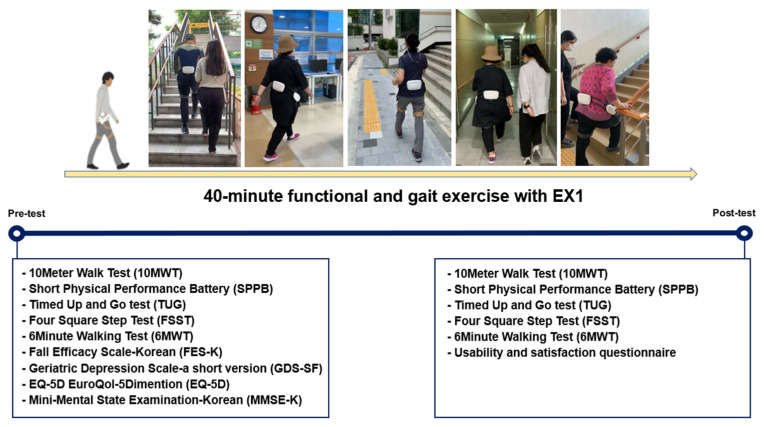
Experimental protocol.

**Figure 2 healthcare-11-00643-f002:**
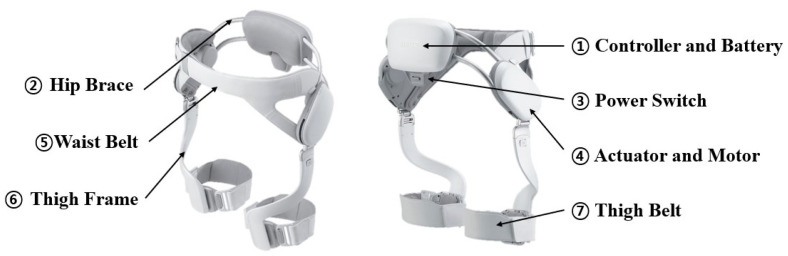
EX1 developed by Samsung Electronics.

**Figure 3 healthcare-11-00643-f003:**
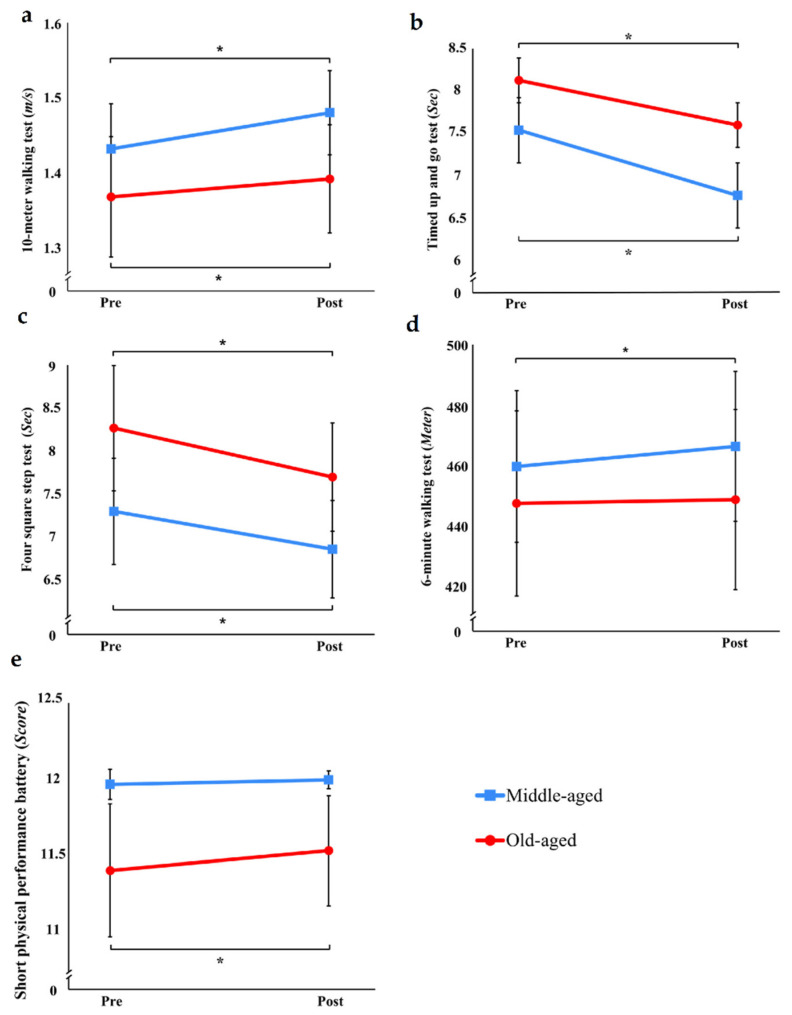
Changes in physical function. (**a**) 10 m walking test; (**b**) timed up and go test; (**c**) four square step test; (**d**) 6 minute walking test; (**e**) short physical performance battery (* *p* < 0.05).

**Figure 4 healthcare-11-00643-f004:**
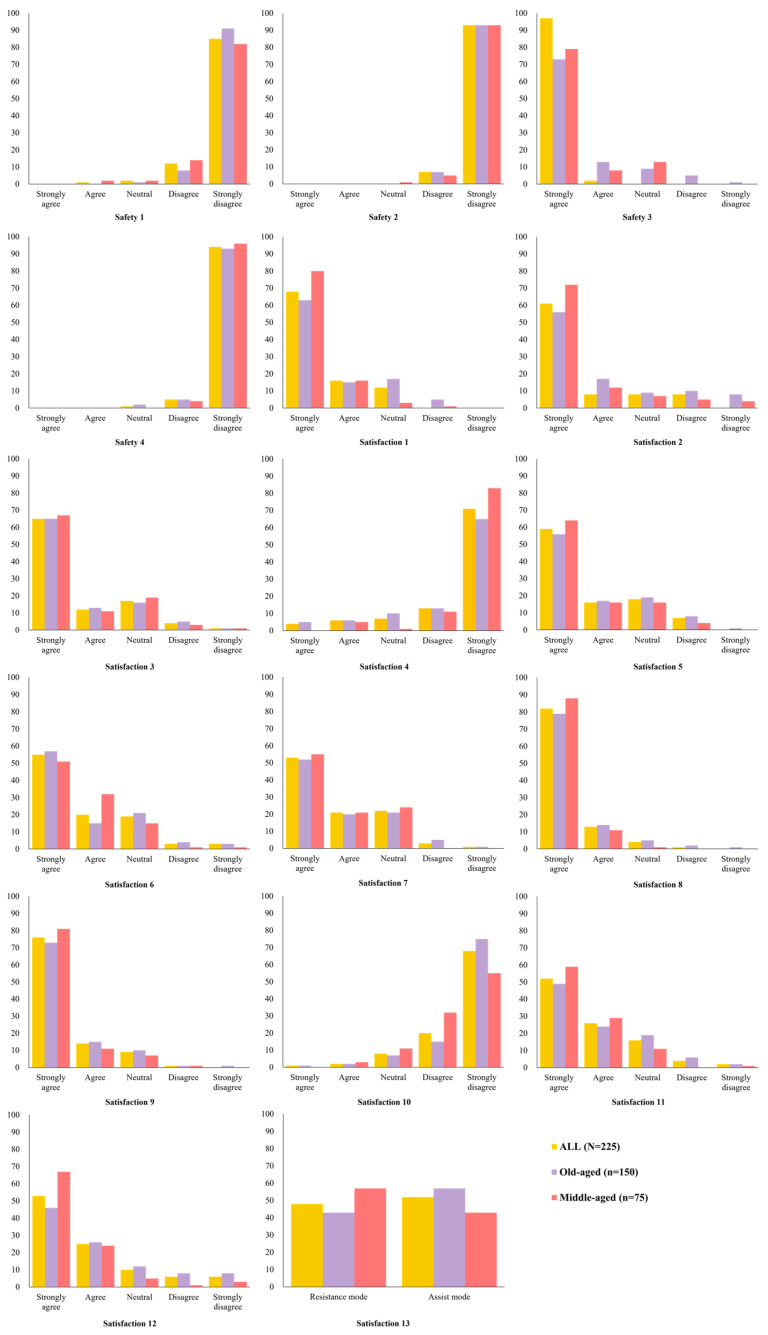
Responses to the questionnaire about usability and satisfaction of the EX1. The items of the usability and satisfaction evaluation include the safety of wearing, the risk of falling, the ease of operation, the feeling of weight, the fit, and the noise of the EX1. Refer to Table 2 for the usability and satisfaction questionnaire.

**Table 1 healthcare-11-00643-t001:** General characteristics of participants.

Characteristics	Total (*N* = 225)	Middle-Aged (*n* = 75)	Old-Aged (*n* = 150)
Gender (male/female)	80/145	31/44	49/101
Age (year)	67.52 (10.99) ^a^	74.31 (4.79)	53.96 (6.30)
Height (cm)	161.53 (8.26)	159.85 (8.85)	164.88 (7.32)
Weight (kg)	61.76 (10.58)	59.93 (9.32)	65.43 (11.98)
BMI (kg/m^2^)	23.59 (3.06)	23.42 (2.92)	23.94 (3.33)
MMSE-K	28.16 (1.82)	27.95 (1.93)	28.57 (1.49)
GDS-15	3.71 (3.32)	4.1 (3.43)	2.93 (2.99)
FES-K	99.32 (3.30)	99.07 (3.92)	99.81 (1.24)
EQ-5D	0.90 (0.08)	0.89 (0.92)	0.93 (0.46)

^a^ Mean (SD), middle-aged group = 40–64 years, old-aged group = 65–84 years; BMI = body mass index; MMSE-K = Mini-Mental State Examination—Korean; GDS-15 = Geriatric Depression Scale-15; FES-K = Fall Efficacy Scale—Korean; EQ-5D = EuroQol-5 dimension.

**Table 2 healthcare-11-00643-t002:** The usability and satisfaction questionnaire of the EX1.

Domain	No.	Item	Strongly Disagree	Disagree	Neutral	Agree	Strongly Agree
Safety	1	Did you have any risk of falling when turning or leaning forward while using the EX1?	①	②	③	④	⑤
Safety	2	Did you have a risk of fall or injury while using the EX1?	①	②	③	④	⑤
Safety	3	Do you think you can control the risks posed by EX1 yourself?	①	②	③	④	⑤
Safety	4	Did you fear falling while walking with the EX1?	①	②	③	④	⑤
Satisfaction	1	Was the EX1 easy to use?	①	②	③	④	⑤
Satisfaction	2	Do you think people with healthy people can also use the EX1?	①	②	③	④	⑤
Satisfaction	3	Were you not shamed for using EX1?	①	②	③	④	⑤
Satisfaction	4	Do you think the EX1 will have negative social perceptions?	①	②	③	④	⑤
Satisfaction	5	Was the weight of the EX1 appropriate for walking?	①	②	③	④	⑤
Satisfaction	6	Was the EX1 comfortable to wear?	①	②	③	④	⑤
Satisfaction	7	Were you satisfied with the material of the EX1?	①	②	③	④	⑤
Satisfaction	8	Do you think the assist mode of the EX1 helps with gait?	①	②	③	④	⑤
Satisfaction	9	Do you think the resistance mode of the EX1 helps with gait exercises?	①	②	③	④	⑤
Satisfaction	10	Was there any disturbance caused by noise when using the EX1?	①	②	③	④	⑤
Satisfaction	11	Are you satisfied with the color and design of the EX1?	①	②	③	④	⑤
Satisfaction	12	Are you willing to continue using the EX1?	①	②	③	④	⑤
Satisfaction	13	Between assist and resistance modes of EX1, which was more helpful?	① Resistance mode ② Assist mode				

## Data Availability

The data used and/or analyzed during the current study are available from the corresponding author on request.

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
