# Peer review of "Functional Improvement and Satisfaction with a Wearable Hip Exoskeleton in Community-Living Adults"

_healthcare, 2023, doi:10.3390/healthcare11050643_

Round 1

Reviewer 1 Report

Recommendation: Minor revisions are needed as noted.

Comments:

This manuscript has demonstrated that a single session of exercise with the EX1 (A personalized robot developed by Samsung Electronics with a lightweight of 2.1kg that is worn on the hip joints) in middle- and old-age persons improved physical performance, including gait and balance, and received positive feedback. A newly developed wearable hip exoskeleton, the EX1, is a potentially useful exercise device for improving gait and physical function not only in the elderly but also in middle-aged people.

Considering a detailed study has been carried out, I have only minor comments addressing experimental design and paper writing, and once these are addressed the work is recommended for publication.

(1) In lines 271-273, the authors claimed that “…although it was not significant after a single session of exercise, walking distance for 6 minutes improved, indicating the potential for positive results with long-term exercise”. In my opinion, this is a very interesting and meaningful point worthy of study, so why the corresponding experiment was not carried out to further verify this conjecture?

(2) It is recommended to optimize Figure 1, as the interior illustrations are not clear enough. Besides, if relevant materials are still retained, it is recommended to add some experimental photos or videos to the Supplementary to better attract readers and intuitively understand.

(3) The abbreviation, such as “EX1” (line 20), “TUG” (line 22), “FSST” (line 23), “6MWT” (line 24), “SPPB” (line 24), and “ICC” (line 128), should be clearly explained when it first appears. In addition, I think some expressions in the manuscript need to be modified, such as “0.68 years” (line 43).

Reviewer 2 Report

despite the interesting topic, the present study has a great weakness.

It not possible to declare a positive effect of a single session of robotic walking; reseachers must observe long term effect of such exoskeleton.

Therefore it is necessary to conduct a more structured study with a specific training protocol repeated in a scheduled program and repeat the experiments pre-post. Consider the paired data t-test for statistical analysis.

Round 2

Reviewer 2 Report

I confirm the previous assessment.

The paper is still declaring (lines 332-338):

"4. Discussion

Our study demonstrates that a single session of exercise with the EX1 improved 232 physical functions. In addition, positive results of the EX1 were confirmed by conducting a usability and satisfaction survey after exercise with EX1. 

In this study, statistically significant improvements in gait speed, balance ability, and gait endurance were confirmed through a single session of exercise with the EX1. The results of this study suggest that a single session of exercise with the EX1 has several key advantages for physical function and efficiency."

In my opinion, the study does not support this conclusion and long term effects need to be investigated.

It is needed a complete study and a more rigorous data analysis.
